# Farmers’ Risk Perception on Climate Change: Transhumance vs. Semi-Intensive Sheep Production Systems in Türkiye

**DOI:** 10.3390/ani12151992

**Published:** 2022-08-06

**Authors:** Sezen Ocak Yetisgin, Hasan Önder, Uğur Şen, Dariusz Piwczyński, Magdalena Kolenda, Beata Sitkowska, Ceyhun Yucel

**Affiliations:** 1Department of Animal Science, Ondokuz Mayis University, Samsun 55139, Türkiye; 2Department of Agricultural Biotechnology, Ondokuz Mayis University, Samsun 55139, Türkiye; 3Department of Animal Biotechnology and Genetic, Faculty of Animal Breeding and Biology, Bydgoszcz University of Science and Technology, 85-796 Bydgoszcz, Poland; 4Organic Agriculture Program, Graduate Education Institute, Taşlıçiftlik Campus, Gaziosmanpaşa University, Tokat 60000, Türkiye

**Keywords:** livestock, sheep, climate change, ‘farmers’ perception, adaptation, Türkiye

## Abstract

**Simple Summary:**

Adaptation strategies developed by sheep breeders against climate change are essential for sustainable production. This study was designed to determine the effects of climate change on perception, the risks of sheep farmers and the actions taken to mitigate these impacts. Nearly all farmers in both production systems agreed on the increased use of drugs and chemicals in their farming activities due to the increase in unknown and known disease outbreaks. The current study showed that transhumance farmers focused mainly on selling livestock as an adaptive strategy. In contrast, semi-intensive farmers concentrated on modifying their farm management and feed operations. Our findings show that semi-intensive farmers do not take deliberated and consistent action against climate change but adapt spontaneously.

**Abstract:**

Sheep farmers’ perceptions of climate change and its impacts and the adaptation strategies they consider to address these risks are of great importance in ensuring the resilience of farming practice. This study focused on sheep farmers’ perception of climate change and the risks and actions taken to mitigate these impacts. A total of 68 surveys were carried out among sheep farmers (39 transhumance and 29 semi-intensive farmers) by two different representative production systems in Türkiye. Variables regarding the socio-economic profile, climate change impacts, and adaptation strategies were identified and analyzed. Principal component analysis and a Pearson Chi-square test were used to evaluate the data. Both farmers’ groups accepted and perceived climate change, showing good awareness and perception. The farmers’ attitudes towards adaptation to climate change were associated with production systems. Transhumance farmers had limited adaptation and coping strategies compared to semi-intensive farmers. Transhumance farmers focused mainly on selling livestock (mostly to cope with degraded natural grassland/feed deficiency) as an adaptive strategy. In contrast, semi-intensive farmers focused on modifying their farm management and feed operations, such as changing the feed ratio and supplement use, improving water and feed storage, and considering crop feed production. The knowledge obtained from this study could be helpful for farmers and policymakers who develop long-term small ruminant production strategies that consider the effects of climate change and adapt them to different farming systems in the Türkiye.

## 1. Introduction

Climate change (CC) is a global fact that threatens our planet with global warming, droughts, flooding, and depleting natural resources [1,2,3]. According to the International Panel of Climate Change (IPCC) forecast, in the next 90 years, the average global temperatures will increase by between 1.8 °C and 4.0 °C [4]. This change will directly impact livestock, farming systems, and human and animal health. Scientific evidence indicates that climate change has already affected the ecosystem [5,6,7,8] in a diverse set of sectors, particularly in agriculture, which is the main source of rural livelihood. It is also a global challenge for humans and their socio-economic activities, health, livelihood, and food security [9,10,11,12]. These negative impacts, combined with socio-demographic pressure, often contribute to food insecurity and poverty, increasing their vulnerability in the end [13]. Climate projections for Mediterranean countries suggest that the region will become warmer and drier with more frequent and extreme weather events [14,15]. In most temperate Mediterranean countries, particularly Türkiye, there is a significant climatic change; excessive drying and warming will occur with less precipitation. According to Turp et al. [16], between 2020 and 2050, temperatures in Türkiye are expected to rise by between 0.5 and 4 °C. This situation is expected to be more severe in the warmer seasons.

Global sustainability policies, such as the Sustainable Development Goals (SDG) by the United Nations (UN), aim to ensure sustainable development with nature conservation and improved human-wellbeing [17]. Many specific SDG objectives are addressed, and their potential contributions to achieving the SDGs are well-covered in the present study, such as responsible production and consumption (SDG-12), climate action (SDG-13), and life on land (SDG-15). Recent reviews of scientific papers demonstrate that climate change, biosphere integrity, and land use are central concerns associated with the planet’s functionality [18].

It is widely recognized that identifying the impacts of climate change and determining the adaptation strategies to combat the effects is crucial to developing realistic and applicable action plans and policies that farmers can adopt [19,20]. Although many farmers dislike the pastoralists’ approach and view their ideas as “irrational” [19], a new trend is emerging among researchers [21,22] who agree that incorporating land users’ perceptions in works would have beneficial intimations on their understanding of the rangeland ecology.

Studies on global warming have generally focused on the ways to reduce the problems that this will cause, based on the modelling of climate data and scenarios. The climate parameters are sufficient to understand climate changes, but the researchers should also determine farmers’ perceptions. Therefore, an in-depth study is needed to examine farmers’ understanding of climate change, its significant impacts on livestock production, and adaptation strategies [23]. The current international scientific consensus indicates that an effective response to climate change requires an understanding of perceptions and observations of individuals [24,25] for applying adaptation strategies [26]. Analyzing farmers’ observations and perceptions of climate change and agricultural production is essential in creating future adaptation strategies and agricultural policies [27,28,29]. It has been determined that individual experience and climate change observations have increased farmers’ intention to adapt [30,31]. Therefore, knowing how farmers perceive climate change and their awareness is critical to deciding on the level of exposure of farmers to this problem and their capacity to adapt. Perceiving the climate change risk is a necessary motivator and a key factor for farmers to start adjusting and determining suitable adaptation strategies [27,32,33]. According to [34], ensuring farmers’ awareness of climate change’s potential risks and impacts would be the essential step towards adaptation. Failure to consider farmers’ perceptions of climate impacts would lead to a loss in developing practical and sustainable strategies.

Türkiye has a long tradition of sheep production, being the 8th biggest sheep-producing country globally, with 35 million people relying on semi-intensive and transhumance systems. Livestock production contributes 22% of its total agricultural value [35], and sheep production has a 17.5% share in total livestock production [36]. They are considered assets with quick returns on the rural livelihood because of their fast multiplication. Transhumance (sheep and goats) in Türkiye is still a dynamic practice closely linked to natural resources and, thus, the climate. As a result, it is one of the most vulnerable communities to climate change, with no current information on adaptation strategies. Recent reviews of scientific studies demonstrate that small ruminants are more sensitive to climate change because of water resources [37] and decreasing pasture quality and quantity [38,39,40,41]. Therefore, the impact of climate change and farmers’ adaptation strategies remains a severe challenge. The continuation of climate change reveals that we need to understand how farmers perceive this problem and how they can develop solutions.

In most cases of semi-intensive sheep breeding (85%), closed pens are used as shelters. In addition to pasture, breeders generally (94%) provide their sheep concentrated feed with varying content in all seasons except winter. In wintertime, sheep feeding is carried out in the barn. The pasture is close to the shelter, and long distances are not covered. The flock usually returns to the shelter in the evening, and supplemental feeding is carried out there. The animals’ water is generally provided in the shelter. Most breeders cultivate field crops for forage, such as barley, wheat and alfalfa. Mating is usually carried out between June and October. Health protection procedures are carried out according to the program planned by the Ministry of Agriculture for the relevant region [42].

Some attempts have been made to analyze farmers’ perception of climate change in pastoral systems, mainly cattle [43,44,45]; however, farmers’ perception of climate change and its impacts on sheep transhumance has not been studied. Limited research on animal production means it is still one of the main issues of interest [44,45,46]. There is little knowledge of transhumance perception and awareness of climate change and how these changes impact sheep production in transhumant communities.

This research will contribute to the literature and is the first of its kind. Based on this background, this paper addresses the following research questions: (i) To what extent do farmers know about climate change (awareness)? (ii) How do they perceive the impacts on their farming? (iii) How do they adapt to the new climate variability/adaptation strategies? 

## 2. Materials and Methods

### 2.1. Study Area and Data Collection

The study was carried out in two districts: Tokat-Erbaa (winter lowland) and Giresun-Karagöl (summer highland) in the Black Sea Region, Türkiye, located in the temperate oceanic climate zone (Figure 1). Both provinces were chosen for this study because Tokat is the winter location and Giresun is the summer settlement area for transhumance communities. Tokat is located at the crossing zone between the central Black Sea and the central Anatolian region of Türkiye, with an elevation ranging 85–2416 m. Geographically, the winter settlement area (Tokat-Erbaa) is located at 40°39′13.05″ N and 36°35′12.67″ E longitudes with an average elevation of 350 m above sea level. Giresun-Karagöl highlands, the summer settlement area, is situated at 40°31′42″ N latitude and 38°10′09” E longitude with an average elevation of 2643 m above sea level and an average annual rainfall of 1288 mm (Figure 1).

The site represents diverse farming systems and apparent differences between the farming communities. The standard livestock unit in this pastoral area is the local Karayaka sheep, well known for its hardiness. The study employed qualitative and quantitative research approaches to collect primary and secondary data at the field level. Prior consent from farmers was obtained before we began the survey, and the research objectives and outcomes were explained to the respondents. Data collection was carried out in October 2020. Two key instruments were used for data collection: a semi-structured survey and focus group discussions organized with various stakeholders in the village. The farms were selected according to the registered farmers’ representatives to the Breeders Association. Interviews were carried out face-to-face by some of the authors. Each survey took about half an hour. The survey was present in Appendix A. In total, 60 transhumance and 92 semi-intensive farms were registered with the Breeders Association of the province. In total, 68 farmers participated. The main topics of the discussion were the impact of climate change on sheep production and the environment and their perceptions of climate change. A total of 68 face-to-face surveys were conducted among farmers (39 transhumance and 29 semi-intensive). Secondary data consisted of long-term meteorological data from nearby weather stations (temperature and precipitation) collected by the Department of Meteorology, Samsun Türkiye.

### 2.2. Data Analysis

The precipitation data calculated the annual rainfall anomaly index (RAI) using the following equations [47] to analyze the frequency and intensity of the dry and rainy years in the studied provinces.
$$\mathrm{RAI}=+3\;\left(\frac{\mathrm{RF}-M_{\mathrm{RF}}}{M_{H10}-M_{\mathrm{RF}}}\right)\;\mathrm{for}\;\mathrm{positive}\;\mathrm{anomalies}$$
$$\mathrm{RAI}=-3\;\left(\frac{\mathrm{RF}-M_{\mathrm{RF}}}{M_{L10}-M_{\mathrm{RF}}}\right)\;\mathrm{for}\;\mathrm{negative}\;\mathrm{anomalies}$$
where RAI is the rainfall anomaly index; RF is the rainfall for the year in question; MRF is the mean actual annual rainfall for the total length of the period; MH_10_ and ML_10_ are the mean of the 10 highest and lowest (respectively) values of rainfall (RF) of the period.

Principal component analysis (PCA) was used on perceptions and adaptations to determine the effective variables that still contain the most information in this large data set. R software (R Core Team, R 4.2.0., Vienna, Austria) was used to analyze the data using *FactoMineR* and *factoextra* libraries. The participants’ responses for the second part of the questionnaire were evaluated with the PCA method to provide more specific validity of their perceptions of climate change. Principal component analysis (PCA) relies on a model of relationships between variables to identify common factors (i.e., the statements in the questionnaire). When these variables are highly correlated, they are effectively “saying the same thing” and are described as components [48]. The obtained factor loadings only show correlations between the derived component score of all individuals and their responses to each questionnaire statement. Significantly contributing variables to the components were selected as influential [49,50,51]. For measurement data, mean, minimum, maximum, frequency, and percentage values are used in descriptive statistics. The data type, description, frequency, and percentage are given if the data are scoring data. While the independent variable is the business type, all the remaining data is considered as the dependent variable. Pearson Chi-Square analysis was used to analyze the relationship between different variable groups. All statistical tests were performed using R software [52].

## 3. Results and Discussion

### 3.1. Socio-Economic Profile of Farmers

The socio-economic profile of the farmers in the study is represented in Table 1. Farming communities are distinguished by characteristics related to household decision-making. Farmers’ ages are associated with the amount of time they spend observing or experiencing climate change while farming, which is an important factor influencing their perceptions. Some studies have taken age as a proxy for farming experience [53], and with more experience, it is assumed that farmers were more likely to have perceived changes in climate [54]. The mean age in households was 46.1 years. All respondents went to formal school, and 50% of respondents completed primary school and above. The choice of adaptation and mitigation techniques is heavily influenced by education. This is because educated individuals are expected to be exposed to better information about climate change, thereby leading to choosing compatible adaptation strategies. Similarly, the education of the farmers is considered vital in their perceptions of climate change [55]. Fifty percent of households had income other than livestock production. All respondents were men (this is due to the fact that we interviewed the head of the family, who in Türkiye is usually a man).

Traditionally small ruminant production in Türkiye is managed mainly by men because of physical labor. A total of 40.3% of respondents’ flocks had 100–200 heads, and 43.3% had more than 200 heads. Since income is a versatile indicator for agricultural resources, it can be associated with the perception of climate change. Non-agricultural activities can be considered a phenomenon that generates additional income and a social mechanism developed by a farmer’s family against climate change [56]. Of these respondents, 73.5% used natural rangelands above 50 ha to graze their animals. Most farmers (57.4%) make decisions about their farms in consultation with their family members, while 32.4% make decisions by themselves. In total, 94.9% of (TH) transhumance respondents were involved in sheep production for longer than 20 years, whereas 51.7% of SI (semi-intensive) farmers were involved for longer than 20 years. A significant number of respondents have heard about “climate change” (91.2%), and 53.1% were using television as a primary information source for the weather forecast. Computers and the internet were pointed to as the primary information source in terms of creating a perception about climate change, and they are evaluated in this context.

### 3.2. Farmer’s Perception on Climate Change Impacts

To determine their level of awareness, farmers were first asked if they had heard of climate change. After this question, they were asked whether they perceived a change in precipitation and temperature in the environment they had lived in for the last 15–20 years. Unsurprisingly, 91.2% of farmers were aware of climate change. Most respondents perceived that climate change is (93.1%) caused by both natural and anthropogenic means. The first step in drafting local adaptation mechanisms is recognizing that human actions are vital in driving global warming trends [57]. Both farming systems perceive climate change. Although farmers are aware of climate change and the contribution of livestock practices to climate change, they lack an understanding of emission production patterns.

Most of the interviewed farmers perceived many changes in climatic factors in their regions over the last 20 years. The transhumance system, which relies entirely on natural resources, is more affected by the impacts of climate change. Most farmers acknowledged a severe rise in temperature and extreme weather events. Farmers perceived an increase in temperature (94.73% of transhumance and 93.1% of semi-intensive system farmers), extreme weather events and disasters such as windstorms (65.79% and 37.93%, respectively), and droughts (54% and 39%, respectively). They also stated that precipitation decreased by 44.8%; however, 49.3% of farmers noticed that it had become “unreliable”.

Farmers noted that “it may not rain for a long period within the rainy season”, and some indicated that “there is unexpected rainfall during the dry season, which is unusual”. Many farmers reported changes in precipitation intensity, onset, and duration. Similar findings have also been reported in the literature [58]. Predominantly, transhumance farmers perceived prolonged drought and unreliable rainfall inflated the incidence of diseases outbreaks that have become severe in the last five years due to climate change. Additionally, they further stated that vaccine and medicine needs had increased by 86.84%. Farmers‘ perceptions and acceptance of climate change are vital because it facilitates the decision-making process for adaptation strategies. Studies have shown that a lack of knowledge about climate change, beliefs and cognitive systems that question the reality of climate change, and resistance to adapt (because there is “no evidence”) are all significant barriers in increasing adaptation ability and preparing farmers for the effects of future climate change [57]. Farmers appear to acknowledge climate change, are open to expanding their knowledge, and are prepared to examine adaptation alternatives, including longer-term plans [12]. Previous studies from Europe, Asia, Africa, and other parts of the world have reported similar findings regarding farmers’ perceptions of rising temperature and decreasing rainfall trends in the past years [12,56,58,59,60,61,62,63,64,65,66,67]. Knowing that almost all the farmers are well aware of climate change, we explored the various perceived risks to their production system.

### 3.3. Determinants of Farmers’ Perceived Climate Change Impacts

Nearly all respondents indicated that their entire production system is negatively affected by climate change.

The PCA results show that most significant livestock production variables for transhumance system determining farmers’ perception of CC were increased use of drugs and chemicals, followed by a decrease in product quality (meat and milk), change in migration route, change in rainfall and decrease in productivity. According to PCA, an increase in extreme events and change in precipitation for semi-intensive farms were influential variables. Variance explanation of PCA was found at 87.71% and 87.99% for TH and SI, respectively.

The contribution of variables of climate change perceptions was identified through PCA. Important variables for the transhumance system were determined as dry season length, followed by wind density in summer, the temperature in summer, and the length of hot and cold periods. The length of hot period, the temperature in the winter, the length of cold period, dry season in summer, and wind density in the dry season and rainy season in winter were effective variables to impact CC perception in the semi-intensive farmers. The variance explanation of PCA was found at 74.43% and 87.10% for TH and SI, respectively.

As for an adaptation strategy, PCA showed that selling livestock was the primary variable for transhumance system. At the same time, diverse feed use, considering crop feed production and more water use, were the primary three variables for SI systems. Variance explanations of PCA were found at 91.09% and 74.12% for TH and SI, respectively.

Table 2 presents climate change’s perceived impact on farming activities. Nearly all respondents stated that their overall productivity had been affected significantly by CC in the last decade (x^2^ = 19,600 *p* = 0,001). In total, 94.73% of TH and 89.66% of SI farmers stated that their productivity decreased. Many transhumance farmers (87.90%) reported that sheep suffer from heat stress in summer. In addition, 93.6% and 81.4% of farmers reported decreased meat production and feed intake (respectively). This is in accordance with other studies from other countries that also reported farmers’ perceptions of productivity loss in past years. The percentage of farmers who associated the increase in temperature with lowered milk production is about 47–73% in Tunisia [28], 41% in Malta [68], 46.7% in Bengal [69], and 56.6% in India [70].

Most SI farmers perceived an increase in labor (68.96%) costs, while transhumance respondents have varying views (x^2^ = 18.87 *p* = 0.003). This variation is associated with the nature of the transhumance system because it relies primarily on family members. Thus, they usually do not recruit workers from the community. A large share of transhumance farmers (67.59%) perceived an increase in natural disasters. Moreover, the majority of (92.11%) noted that they had to change their migration routes between winter and summer pastures mainly due to the negative impact of climate change on water and natural rangeland resources. Nearly all farmers in both production systems noted the increased use of drugs and chemicals in their farming activities due to the increase in unknown and known disease outbreaks. Farmers in both systems stated that prolonged drought and delays in rainfall/unreliable rainfall due to climate change affected the quality and availability of rangelands and grasslands.

### 3.4. Farmers’ Adaptation Strategies to Climate Change

Adaptation strategies adopted by different production systems mostly involved changes in feeding and management practices. We asked the farmers, who stated that they had observed changes in the climate in the last 20 years, to indicate the adjustments they made in their farming activities (Table 3). A remarkable determination to adapt to climate change was observed in semi-intensive farmers. Almost all SI farmers had adapted their practices in response to changing climate conditions. Most had adjusted their management of farming resources to climate change with biophysical and farm operational adaptation measures. They optimized feed/ratio (57.14%) and feed storage (82.76%), added supplements to their ration (68.7%), stored water (79.31%), and sold livestock (70.1%). These findings are consistent with those of Hamadeh et al. [71] and Demirbuk [67]. Increased reliance on feed supplements has also been noted in response to feeding shortages during periods of drought when pastures are inadequate [72,73]. Surprisingly, many pastoral farmers have adopted strategies such as converting to crop feed production (73.68%) and selling livestock to sustain their farming (63.8%).

According to the available feed resources, farmers determine flock size and develop new adaptation strategies that focus on management practices [71,73]. The reduction in flock size in both transhumance and semi-intensive systems responds to increased production costs. This rather dramatic strategy can represent a short-term response to circumstances or a long-term livelihood strategy. In addition, this situation shows that small cattle are a versatile investment tool that acts as emergency insurance [74]. Farmers address these difficulties by developing an effective strategy that brings solutions to their problems.

All farmers in both groups answered “No” when asked if they considered changing their breeds, reasoning that they know the breed’s hardiness well. If we compare the two production systems, there is a clear difference in the uptake of adaptation strategies. We have investigated why transhumance farmers behave in such an inattentive and submissive manner. One of the main reasons we noted during our group discussions was their strong feelings of being neglected by the community and the administration. The farmers’ first constraint was the lack of motivation to sustain sheep production, caused mainly by lowered availability and the poor quality of the natural rangelands/pastures. The agricultural expansions greatly limited access to the natural rangelands and agricultural lands and increased pasture rental fees.

It is worth noting that many transhumance respondents want to implement adaptation strategies but do not know how to do so (57.1%). Additionally, the lack of cash flow and extension service availability (the fact that they are in a remote area) are other reasons they do not concern themselves much with mitigation strategies. As considered by farmers, the main barriers to climate change adaptation were lack of knowledge (57.1%) and lack of financial resources (46.9%). The results were consistent with other scientists’ results [75,76,77,78,79,80]. Governments should provide financial incentives to increase farmers’ adaptability and reduce financial risks [76].

The first crucial step for adaptation success is to change farmers’ attitudes and increase their awareness of climate change [81,82]. It is also important to educate them that risk and apprehension significantly influence adaptation behavior [83,84]. At the same time, Osberghaus et al. [85] argued that risk perception is insufficient to promote adaptation. Risk perception and concern are an important part of adaptation motivation but not a sole influencing factor [76,85,86].

Our findings have confirmed that the most important determinant of adaptation is the risk perceptions of various climatic events on agricultural production, which is also consistent with the findings of other authors [45,83]. Our results show that SI farmers do not take deliberated and consistent action against climate change but adapt spontaneously [87]. These relatively inexpensive adaptation measures include reactive operational and biophysical adjustments in the farms, e.g., changing feed, improving feed storage, selling the animal to adjust feed availability, and using water storage. This shift in the source of income is an example of advanced risk management. It shows that farmers’ adaptive responses combine economic incentives to sustain their livelihoods and previous experience with the negative impacts of climate events. Farmers’ life experience is no guarantee of successful adaptation, but adaptation can delay or reduce these adverse effects [88,89]. More specific interventions to address these perceived risks in the production chain need to be incorporated into adaptations targeting farmers from different demographic, socio-economic, and institutional backgrounds [45].

### 3.5. Comparing Farmers’ Perceptions with the Meteorological Data

The rainfall anomaly index (RAI) for the Tokat (from 1990 to 2018) and Giresun provinces (from 1990 to 2013) are presented in Figure 2 and Figure 3, respectively. Classification of RAI intensity was made according to Costa et al. [90]. The bars with positive values represent rainy years at different degrees of rain intensity, and the bars with negative values represent dry years with varying degrees of intensity. Eleven years with a positive RAI, changing from highly wet to humid, and seventeen years with a negative RAI, ranging between very dry and dry, were observed in Tokat province. In Giresun province, on the other hand, positive RAI varying from highly wet to humid in seven years and negative RAI varying from highly wet to moist in fourteen years were observed. In other words, both provinces had more years of drought than rainy ones.

In the present study, most of the interviewed farmers stated that there had been various changes in the climatic factors in the region in the last 20 years. In particular, the transhumance farmers noted that meteorological anomalies had increased in the previous five years, and the incidence of prolonged drought and unreliable rainfall due to climate change had increased. These arguments are also supported by RAI indices derived from meteorological precipitation data of both Tokat and Giresun provinces. In Tokat province, extreme precipitation was observed between 2008 and 2012, while excessive droughts occurred between 2013 and 2018. Interestingly, while an acceptable drought occurred in Giresun province between 1990 and 2002, a one-year-precipitation–one-year-drought trend was observed from 2003 to 2013. Some dry periods followed the rainy seasons, and a marked dry season may be the reason for the remarkable variability in precipitation in recent years, which may be evidence of changes in the climate [22,91].

## 4. Conclusions

This study highlighted the farmers’ perception of climate change. It provided insight into how farmers in different systems perceive the risks associated with climate change and what adaptation strategies they are considering. We found very high awareness of the relationship between extreme weather events and climate change among semi-intensive and transhumance farmers. Almost all farmers observed an increase in the frequency and severity of extreme climatic events such as drought, heat, and unreliable rainfall, reflecting actual trends in rainfall and temperature in the study area. It is widely known that awareness and acceptance of the risks associated with climate change help make adaptation decisions. Although most transhumant farmers (active observers of climate and ecosystem changes) recognized the risks associated with climate change, they were not diligent in considering adaptation strategies other than animal sales to sustain their farming. This approach further contributed to their low adaptive capacity. While animal sale seems to be a remedy, the risk of decreasing livestock production and food security is disregarded. The primary issue with this inattentive approach is the risk of the future sustainability of the transhumance production system. Conversely, semi-intensive farmers primarily sought better feed and improved water storage to adapt their farming operations. Other identified strategies related to feed and supplement use were also considered.

To motivate farmers to develop an adaptive capacity, some actions should be made. It is important to ensure that strategies and actions that are appropriate for animal production systems are included in policy makers’ goals and the associated risks. We strongly suggest developing regionally focused and stakeholder-tailored recommendations to improve understanding of climate change and develop climate change adaptation strategies at a regional level. Designing production-system-specific early-warning policy systems and an adaptive strategy would be long-term realistic adaptation measures.

## Figures and Tables

**Figure 1 animals-12-01992-f001:**
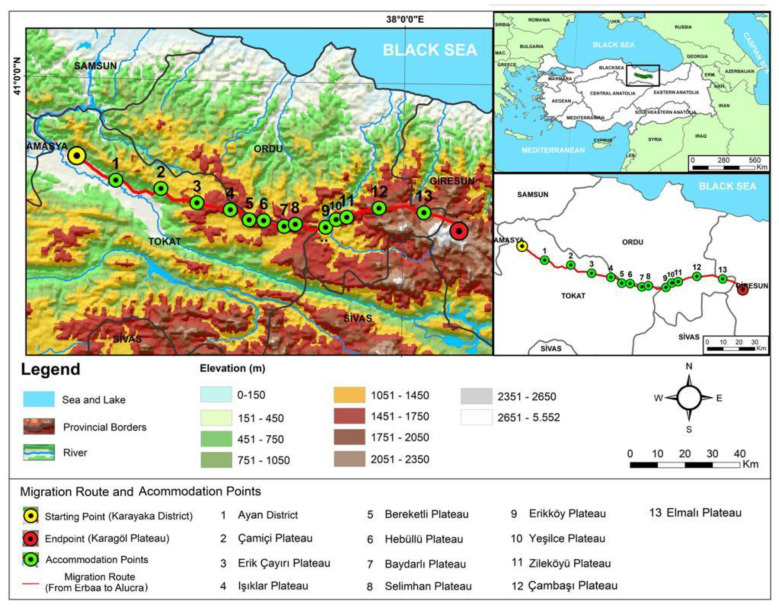
Study area location with migration route.

**Figure 2 animals-12-01992-f002:**
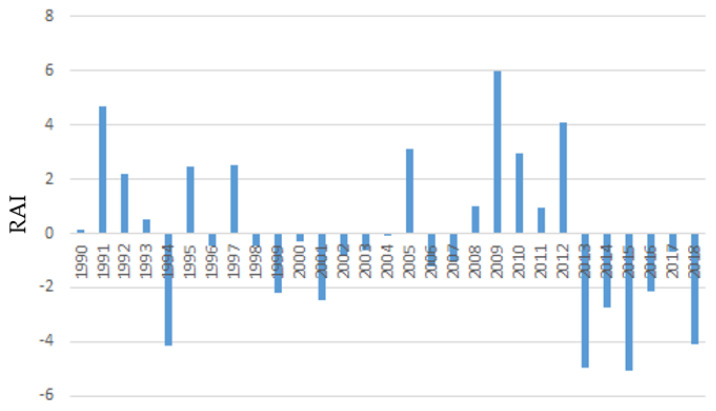
Rainfall anomaly index (RAI) for the Tokat province.

**Figure 3 animals-12-01992-f003:**
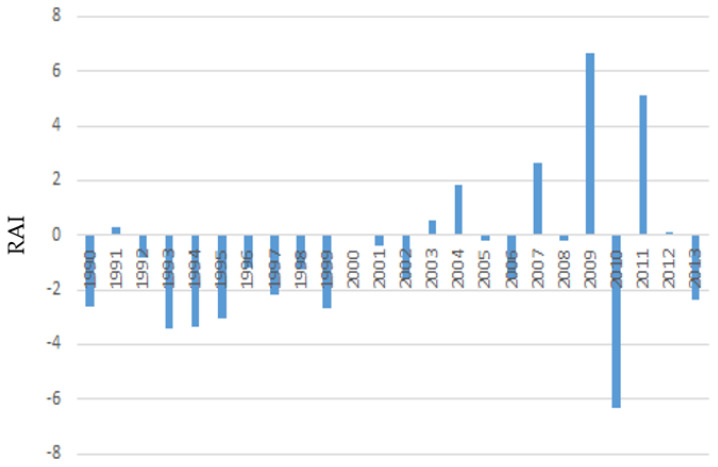
Rainfall anomaly index (RAI) for the Giresun province.

**Table 1 animals-12-01992-t001:** Socio-economics of the farmers’ profiles.

Summary Variables	Description	Percent (%)
Production system	Transhumance	57.4
	Semi-intensive	42.6
Education	illiterate	3.0
	Primary school	50.0
	Secondary school	22.1
	High school	14.7
	College	10.3
How long had they been working in agriculture?	<10	18.7
	>10	81.3
Do they have non-agricultural income?	No	50.0
	Yes	50.0
Do children work on their farm?	No	76.1
	Yes	23.9
Flock size (heads)	<100	16.4
	100–200	40.3
	>200	43.3
Size of grazing land (ha)	<20	20.6
	20–50	5.9
	>50	73.5
Role in the decision-making process	Alone	32.4
	With family	57.4
	Board of directors	10.3
Knowledge of climate change	No	8.8
	Yes	91.2
Sources of climate information	TV	53.1
	Internet	7.8
	TV and internet	39.1
	Mean	Std. Dev.
Age (years)	46.1	11.8
Household size (number)	7.1	4.5

**Table 2 animals-12-01992-t002:** Perceived impact of climate change according to farmers.

	Transhumance (Number of Farmers)					Semi-Intensive System (Number of Farmers)				
Scale *	1	2	3	4	5	1	2	3	4	5
Decrease in productivity	1		1	21	15			1	2	26
Increase in labor cost	15	7	6	8	2	5		4	10	10
Change in temperature	1		2	19	16				2	27
Incidence of natural disasters	8	5		14	11	7	5	6	2	9
Change of migration route		2	1	14	21	NA				
Increase in use of drugs and chemicals	2	2	1	22	11	1		1		27
Increase in water consumption	14	4	2	14	4	10		7	3	9
Difficulties in paying loans	14	2	5	14	3			7	3	19
Decrease in natural rangeland/grassland	3	3	1	17	14		1		6	22

*: (1) Strongly disagree; (2) Disagree; (3) No idea; (4) Agree; (5) Strongly agree; NA: Not Avaliable.

**Table 3 animals-12-01992-t003:** Farmers’ adaptation and coping strategies.

	Transhumance (TR)		Semi-Intensive (SI)	
	Yes (%)	No (%)	Yes (%)	No (%)
Change in feed/ratio	23.68	76.32	57.14	42.86
Supplement usage	10.68	89.32	68.70	31.30
Improved water storage	50.00	50.00	79.31	20.69
Improved feed storage	31.58	68.42	82.76	17.24
Diverse feed usage	26.20	73.80	50.70	49.30
Selling livestock	63.80	36.20	70.10	29.90
Rainwater harvesting		100.00	1.50	98.50
Considering crop feed production	73.68	26.32	24.14	75.86

## Data Availability

The data can be reached to applying the corresponding author.

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
