# Peer review of "Farmers’ Risk Perception on Climate Change: Transhumance vs. Semi-Intensive Sheep Production Systems in Türkiye"

_animals, 2022, doi:10.3390/ani12151992_

Round 1
Reviewer 1 Report
The manuscript addresses farmers' risk perception on climate change and involves in-depth interviews.
Abstract suggest that some of the significant data from the study be included in the abstract. For instance data from Table 3 shows limited adaptation and coping strategies by transhumance farmers compared to semi-intensive. Line 33 farmers focused on "ways" to modify their farm
Introduction
The abstract defined transhumance and semi-intensive sheep production systems. However, in lines 98 to 105 more details could be explained about the practices applied and additional paragraph should address some of the practices of semi-intensive sheep production.
Materials and Methods
It is suggested that a table with the survey questions and another table with some of the key focus group questions be added to improve content of the methods section. There is little information about specific questions that were asked. In addition, who how was the survey sent to farmers, who served as the moderator for the focus group, and who served as the interviewer for the face to face interviews. More details need to be provided to be able to understand and/or repeat the study.
Results and Discussion
Line 196 Should it be all respondents were male or men because we interviewed with the head of the household, who is usually a man.
Line 205-205 Should it be make decisions instead of take decisions?
Line 217 Should it be 91.2% of the farmers interviewed had heard and were aware of climate change?
Table 1. Typo College
Under Whether children work in farm change to Yes
Table 2 Unclear what the Scale 1 to 5 means, good to explain in a footnote.
Line 317 Change to These findings
Conclusions
Line 429 Should it be adaptive strategies would be long-term realistic adaption measures. Check sentence for clarity.
Author Response
For Reviewer 1
Reviewer Comment
Abstract suggest that some of the significant data from the study be included in the abstract. For instance data from Table 3 shows limited adaptation and coping strategies by transhumance farmers compared to semi-intensive. Line 33 farmers focused on "ways" to modify their farm
Answer
The sentence revised adding the expression “change in feed/ratio, supplement use, improved water storage, better feed storage and considering crop feed production.” Please see lines 31-32 and 35-36 in revised manuscript.
Reviewer Comment
The abstract defined transhumance and semi-intensive sheep production systems. However, in lines 98 to 105 more details could be explained about the practices applied and additional paragraph should address some of the practices of semi-intensive sheep production.
Answer
The paragraph was added
“In most of the semi-intensive sheep breeding (85%) closed pens are used as shelters. In addition to pasture, breeders generally (94%) give their sheep concentrated feed with varying content in all seasons except winter. In winter time sheep feeding is doing in the barn. The pasture is close to the shelter and long distances are not covered. The flock usually returns to the shelter in the evening and supplemental feeding is done in the shelter. The access of animals to water is usually provided in the shelter. Most of the breeders cultivate field crops for forage purposes such as barley, wheat and alfalfa. Mating is done generally between June and October. Health protection procedures are carried out according to the program planned by the Ministry of Agriculture for the relevant region [42].” Please see lines 109-118 in revised manuscript.
Reviewer Comment
It is suggested that a table with the survey questions and another table with some of the key focus group questions be added to improve content of the methods section. There is little information about specific questions that were asked. In addition, who how was the survey sent to farmers, who served as the moderator for the focus group, and who served as the interviewer for the face to face interviews. More details need to be provided to be able to understand and/or repeat the study.
Answer
The expression was added as “Interviews were done by the authors by face to face. Each survey took about half an hour. The survey was given in Supplementary Table 1.” Please see lines 154-157 in revised manuscript.
Reviewer Comment
Line 196 Should it be all respondents were male or men because we interviewed with the head of the household, who is usually a man.
Answer
Correction was done as “All respondents were male or men because we interviewed with the head of the house-hold, who is usually a man.” Please see lines 211-213 in revised manuscript.
Reviewer Comment
Line 205-205 Should it be make decisions instead of take decisions?
Answer
Correction was done as “make decisions” Please see line 221 in revised manuscript.
Reviewer Comment
Line 217 Should it be 91.2% of the farmers interviewed had heard and were aware of climate change?
Answer
Correction was done as “farmers interviewed had heard and were aware of climate change” Please see line 232 in revised manuscript.
Reviewer Comment
Table 1. Typo College
Answer
Correction was done as “College” Please see line 248 in revised manuscript.
Reviewer Comment
Under Whether children work in farm change to Yes
Answer
Correction was done as “Yes” Please see line 248 in revised manuscript.
Reviewer Comment
Table 2 Unclear what the Scale 1 to 5 means, good to explain in a footnote.
Answer
Footnote was added as “(1) Strongly disagree; (2) Disagree; (3) No idea; (4) Agree; (5) Strongly agree.” Please see lines 321-322 in revised manuscript.
Reviewer Comment
Line 317 Change to These findings
Answer
Correction was done as “findings” Please see line 333 in revised manuscript.
Reviewer Comment
Line 429 Should it be adaptive strategies would be long-term realistic adaption measures. Check sentence for clarity.
Answer
Correction was done as “adaptive strategies would be long-term realistic adaption measures.”

Reviewer 2 Report
In my opinion, this is a scientific work that unequivocally adds another valuable brick to the great tower of knowledge that needs to be built to comprehensively understand the effects of global warming on populations and their livestock. Works of this nature will certainly help in the development of strategies for man's adaptation to climate change since strategies to mitigate the effects of global warming on different ecosystems have not yet started to be applied on a global scale and even when this is done the positive effects will take time to appear. Finally, strategies for dealing with the problem, such as the change in animal husbandry habits by farmers who raise sheep in a semi-intensive regime, as the work shows, will end up being increasingly important to maintain the supply of products of animal origin. and also the way of life of this type of economic activity.
Author Response
We thank the referee for his/her valuable comments.

Reviewer 3 Report
This study presents sheep farmers' perceptions concerning the impact of climate change on the sustainability of the sector in Turkey. The subject is modern, interesting and important from academic and policy points of view and relevant to the scope of the Journal. However, the manuscript is not well written, and I think that the paper could not be potentially considered for publication (reject).
Overall, the paper is fairly organized, but the use of English is not good. This is one the main drawback of the manuscript; the manuscript should be revised by a native speaker. There are many spelling and grammatical errors and, in some points, (see for example paragraph in lines 66-72) the text is confusing and must be rephrased.
The other major drawback of the paper is the application of the statistical analysis. The authors state that they apply Pearson chi-square (I assume for testing independence to determine whether two categorical or nominal variables are likely to be related or not). The presentation of the results concerning this analysis (l. 278-300) is very confusing and I cannot understand which variables are included in the contingency tables for testing independence. This makes me believe that the authors are not very familiar with the applied methodology. They refer to measurement data (I assume they continuous data) and scoring data (I assume they mean nominal or ordinal data) and state (l. 175) that for the former they estimate frequency values and percentages. If these values have been transformed into categorical variables the authors should elaborate on this and explain how and why.
Author Response
For Reviewer 3
Reviewer Comment
This study presents sheep farmers' perceptions concerning the impact of climate change on the sustainability of the sector in Turkey. The subject is modern, interesting and important from academic and policy points of view and relevant to the scope of the Journal. However, the manuscript is not well written, and I think that the paper could not be potentially considered for publication (reject).
Answer
Even the valuable reviewer reject the manuscript we managed to revise it according to his/her comments.
Reviewer Comment
Overall, the paper is fairly organized, but the use of English is not good. This is one the main drawback of the manuscript; the manuscript should be revised by a native speaker. There are many spelling and grammatical errors and, in some points, (see for example paragraph in lines 66-72) the text is confusing and must be rephrased.
Answer
Indeed, this manuscript was checked by native speaker. Some changes were made. Please see line 74 in revised manuscript.
Reviewer Comment
The other major drawback of the paper is the application of the statistical analysis. The authors state that they apply Pearson chi-square (I assume for testing independence to determine whether two categorical or nominal variables are likely to be related or not). The presentation of the results concerning this analysis (l. 278-300) is very confusing and I cannot understand which variables are included in the contingency tables for testing independence. This makes me believe that the authors are not very familiar with the applied methodology. They refer to measurement data (I assume they continuous data) and scoring data (I assume they mean nominal or ordinal data) and state (l. 175) that for the former they estimate frequency values and percentages. If these values have been transformed into categorical variables the authors should elaborate on this and explain how and why.
Answer
For this survey study Pearson chi-square method was used for statistical analysis of the relationship between different variable groups after effective variables determined by PCA.
The reviewer emphasize that “This makes me believe that the authors are not very familiar with the applied methodology”. The corresponding author (Hasan Önder) of this manuscript working on biometry and biostatistics and he is a well-known and respected scientist in his field. There is no erroneous or wrong statistical method in this manuscript which have such a simple statistical procedures.
The authors read the paragraphs mentioned by the referee and they detected no miswritten or confusing sentence.
Round 2
Reviewer 1 Report
Line 111 Change... In winter time sheep feeding is done in the
Line154 Change Each survey spent took about half an hour. The survey text is given in Supplementary Table 1.
Author Response
Reviewer Comment
Line 111 Change “doing” to “done”
Answer
The correction was made. Please look at line 120 in the text.
Reviewer Comment
Line154 Change Each survey spent took about half an hour. The survey text is given in Supplementary Table 1.
Answer
The reviewer corrected the sentence in the text. Please look at lines 166-167 in the text.
Thank you for your valuable efforts.

Reviewer 3 Report
The authors do not seem to have taken into consideration the recommendations concerning the use of English in the manuscript, which in my opinion constitutes the major drawback of this paper. The use of English language is not good and rfr this reason I believe that the manuscript is not suitable for publication.
Author Response
Reviewer Comment
The authors do not seem to have taken into consideration the recommendations concerning the use of English in the manuscript, which in my opinion constitutes the major drawback of this paper. The use of English language is not good and rfr this reason I believe that the manuscript is not suitable for publication.
Answer
The language was checked and corrected by Assist. Prof. Dr. Betül ÖZCAN DOST is a staff member of the Translation and Interpreting department.
Thank you for your valuable comments.
